Description of Arundel Clay ornithomimosaur material and a reinterpretation of Nedcolbertia justinhofmanni as an “Ostrich Dinosaur”: biogeographic implications

Brownstein Chase Doran chasethedinosaur@gmail.com
Stamford Museum and Nature Center , Stamford, CT , USA
Wedel Mathew
Electronic publication date: 2017 Mar 8
Publication date: 2017
Volume: 5
Electronic Location ID: e3110
Received 2016 Jul 21; Accepted 2017 Feb 20
Copyright: © 2017 Brownstein
Copyright year: 2017
Copyright holder: Brownstein
License: This is an open access article distributed under the terms of the Creative Commons Attribution License, which permits unrestricted use, distribution, reproduction and adaptation in any medium and for any purpose provided that it is properly attributed. For attribution, the original author(s), title, publication source (PeerJ) and either DOI or URL of the article must be cited.
License URL: https://creativecommons.org/licenses/by/4.0/

Keywords: Ornithomimosaurs, Early Cretaceous, Maryland, Arundel formation, Potomac group, Theropods, Biogeography

Funding: The authors received no funding for this work.

==============================
The fossil record of dinosaurs from the Early Cretaceous of Eastern North America is scant, especially since a few stratigraphic units from the east are fossiliferous. Among these stratigraphic units, the Arundel Clay of the eastern seaboard has produced the best-characterized dinosaur faunas known from the Early Cretaceous of Eastern North America. The diverse dinosaur fauna of the Arundel Clay has been thoroughly discussed previously, but a few of the dinosaur species originally described from the Arundel Clay are still regarded as valid genera. Much of the Arundel material is in need of review and redescription. Among the fossils of dinosaurs from this stratigraphic unit are those referred to ornithomimosaurs. Here, the researcher describes ornithomimosaur remains from the Arundel Clay of Prince George’s County, Maryland which may be from two distinct ornithomimosaur taxa. These remains provide key information on the theropods of the Early Cretaceous of Eastern North America. Recent discoveries of small theropod material from the Arundel Clay possibly belonging to ornithomimosaurs are also reviewed and described for the first time. The description of the Arundel material herein along with recent discoveries of basal ornithomimosaurs in the past 15 years has allowed for comparisons with the coelurosaur Nedcolbertia justinhofmanni, suggesting the latter animal was a basal ornithomimosaur rather than a “generalized” coelurosaur as it was originally described. Comparisons between the Arundel ornithomimosaur material and similar Asian and European specimens suggest that both extremely basal ornithomimosaurs and more intermediate or derived forms may have coexisted throughout the northern hemisphere during the Early Cretaceous.

Introduction

The fossil record of dinosaurs from Eastern North America during the Cretaceous is sparse compared that of the west of the continent, and the most well-known dinosaur fauna from Eastern North America (on the basis of number of taxa and specimens) comes from the Arundel Clay of the Potomac Group of Maryland. This unit, which is Aptian in age (Kranz, 1998), has yielded specimens of the sauropod dinosaur Astrodon johnstoni, the ornithopod Tenontosaurus sp., the nodosaur Priconodon crassus, ceratopsian material (Kranz, 1998; Weishampel et al., 2004; Weishampel, 2006). A number of theropods have also been described from the Arundel, including Deinonychus sp. and the dubious taxa Allosaurus medius, Creosaurus potens, and Coelurus gracilis.

Some of the first ornithomimosaur material to come from the Arundel Clay has been the subject of taxonomic confusion (Weishampel, 2006). Originally found by Lull (1911) to be the bones of an ornithopod he named Dryosaurus grandis, the ornithomimosaur specimens from the Arundel have been described as a species of Ornithomimus (“O.” affinis), referred to the genus Archaeornithomimus, regarded as a small theropod of indeterminate affinities, and finally regarded as an ornithomimosaur of indeterminate affinities (Gilmore, 1920; Russell, 1972; Smith & Galton, 1990; Makovicky, Kobayashi & Currie, 2004; Weishampel, 2006). The metatarsal III and pedal ungual described by Gilmore (1920) were also referred to as Ornithomimus sp. or Ornithomimus affinis by Serrano-Brañas et al. (2016). However, if any of the material were to be determined a distinct and valid species in the future, the original species name (grandis) given by Lull (1911) would take priority. Given the rather unlikely inclusion of the Arundel material into the Late Cretaceous western taxon Ornithomimus as a distinct species, the proper name would indeed be O. affinis due to the preoccupation of Ornithomimus grandis by “O.” grandis Marsh (1890).

Gilmore (1920) originally described the Arundel material as then a new species of ornithomimosaur based on some pedal elements and caudal vertebrae. Additionally, Weishampel & Young (1996) noted that pedal elements and the proximal portion of a tibia were retrieved in 1992. These are in the collections of the United States National Museum of Natural History. Recently, an astragalus was recovered from Prince George’s County, Maryland in 2010 (USNM PAL 540727). Additional material has also been recovered recently from the Arundel but never has been formally described until now.

Early Cretaceous ornithomimosaur remains have been retrieved from Western North America (Ostrom, 1970; Galton & Jensen, 1975), Europe (Sanz & Wenz, 1988; Perez-Moreno et al., 1994; Naish, 2011; Neraudeau & Allain, 2012; Allain et al., 2014), Asia (Maleev, 1954; Dmitiriev, 1960; Kalandadze & Kurzanov, 1974; Xu & Wang, 1999; Boonchai, Grote & Jintasakul, 2009; Ji et al., 2003; Molnar & Obata, 2009; Buffetaut, Suteethorn & Tong, 2009; Makovicky et al., 2009; Jin, Chen & Godefroit, 2012), and Africa (Choiniere, Forster & de Klerk, 2012). The rich fossil record of Early Cretaceous ornithomimosaurs, which has developed in the past decade has allowed for comparisons of the Arundel specimens with a multitude of new taxa.

Here, I describe new specimens of Arundel ornithomimosaurs in the collections of the Dinosaur Park office at Mount Calvert Historical Park in Upper Marlboro, Maryland. These specimens, which were discovered isolated at the Dinosaur Park site, include two different morphotypes of pedal ungual that indicate the presence of two different ornithomimosaur genera within the Arundel Clay. The new Arundel ornithomimosaur material has implications for the evolution of more derived members of the Ornithomimosauria, suggesting that they were present across North America during the Early Cretaceous. However, the lack of specimens able to be assigned to a single animal means that any naming of a new Arundel taxon or new taxa must wait until a skeleton is recovered which can be confidently thought of as formed by associated material. Additionally, a reinterpretation of Nedcolbertia justinhofmanni from the Early Cretaceous of Utah as an ornithomimosaur is provided. The biogeographic and ecological implications of more basal and more derived ornithomimosaurs coexisting in North America are discussed, though the paucity of material from these North American forms during the Early Cretaceous makes any conclusions limited. The basal and derived features found in the Arundel ornithomimosaur material may be indicative that both basal and more derived ornithomimosaurs existed in the Arundel. The coexistence of two Early Cretaceous ornithomimosaur genera is also seen in the Early Cretaceous Yixian Formation of China (Ji et al., 2003; Jin, Chen & Godefroit, 2012).

Methods

Permits

No permits were required for the described study, which complied with all relevant regulations. Access to the collections of the Dinosaur Park office, Upper Marlboro, Maryland was given by Mr. Benjamin Miller.

Institutional abbreviations

I use the term NHRD-AP to refer to the National and Historical Resources Division Archaeology Program collections of fossils from Dinosaur Park, Maryland. I use the term USNM PAL and USNM V to refer to the paleontology collections of the United States National Museum of Natural History, Washington, DC.

The specimens described herein were photographed using a Canon Powershot G-12 digital camera and cropped for figures using Apple Preview.

Results

Geological setting

The Arundel Clay is made up of black lignite and limonite and siderite massive dark-gray mudstones, appearing as discontinuing elongate sediments probably formed as deposits from oxbow swamps (Brenner, 1963; Kranz, 1998; Lipka et al., 2006). It has been debated whether the sediments attributed to the Arundel comprise a distinct formation, a member of the Patuxent Formation, or a member of the Potomac Group alongside a Patuxent member (Kranz, 1998; Lipka et al., 2006; Stanford et al., 2010). Here, the classification of Lipka et al. (2006) referring to the Arundel Clay as a member of the Potomac Formation is followed. The sediments referred to as the Arundel are Aptian in age and have produced a diverse vertebrate fauna, including saurischian and ornithischian dinosaurs, testudines, anurans, the shark Hybodus, and the lungfish Ceratodus (Kranz, 1998; Weishampel et al., 2004; Weishampel, 2006). Each of the fossils which are for the first time described herein (NHRD-AP 2015.v.103.9, NHRD-AP 2014.s.196, NHRD-AP 2016.5.503, NHRD-AP 2014.s.195, NHRD-AP 2014.s.197, NHRD-AP 2014.s.198, USNM PAL 529423 (cast), and NHRD-AP 2016.v.1104) was found isolated at the Dinosaur Park site in Maryland. Kranz (2004) established this site as the locality where the holotype of the sauropod A. johnstoni was found, remarking on the fact that the discovery of the holotype Astrodon teeth had originally been said to have been made near Bladensburg. Kranz (2004) explains that, in actually, the site of discovery of these teeth is near Muirkirk. Importantly, Gilmore (1920) remarked that the majority of the ornithomimosaur remains he described as “Ornithomimus” affinis were collected near Muirkirk. However, Gilmore (1920) does not state the precise site of collection. Thus, it may be that some of the originally described ornithomimosaur material came from nearby sites to the Dinosaur Park site from which the material described herein was collected.

Systematic paleontology

Dinosauria Owen, 1842

Theropoda Marsh, 1881

Ornithomimosauria Barsbold, 1976

Ornithomimosauria indet.

Material: NHRD-AP 2015.v.103.9, proximal and distal ends of a humerus; NHRD-AP 2014.s.196, manual ungual; NHRD-AP 2016.5.503, caudal vertebra; NHRD-AP 2014.s.195, NHRD-AP 2014.s.197, NHRD-AP 2014.s.198, USNM PAL 529423 (cast), NHRD-AP 2016.v.1104, pedal unguals.

Description: The eroded proximal and distal portions of a small left humerus, NHRD-AP 2015.v.103.9 were preserved (Figs. 1A–1F). The humerus is hollow, and along with the size of the specimen, this feature suggests the humerus came from a small to medium-sized theropod dinosaur. The proximal end measures 101 mm long proximodistally, while the distal end is 70 mm in proximodistal length (Table 1). The humeral head is well preserved. The deltopectoral crest is eroded, but does not seem to have been very prominent as there is no indication of any significant raised portion of bone on the portion of the shaft to which the deltopectoral crest corresponds. As in ornithomimosaurs, the preserved portion of the shaft is relatively straight (Makovicky, Kobayashi & Currie, 2004). The distal portion of the humerus is badly eroded, and the distal condyles are almost completely worn way. The humerus is most similar to that of Harpymimus okladnikovi in its relatively robust nature and size of its distal condyles relative to the proximal end (Fig. 6.4E in Makovicky, Kobayashi & Currie, 2004). Because dromaeosaurids (e.g., Deinonychus), troodontids (e.g., Geminiraptor), oviraptorosaurs (e.g., Microvenator), and therizinosaurs (e.g., Falcarius) are also known from the Arundel Clay and other North American units of similar Early Cretaceous age (Weishampel & Young, 1996; Lipka, 1998; Weishampel et al., 2004; Kirkland et al., 2005; Weishampel, 2006; Senter et al., 2010; Senter, Kirkland & Deblieux, 2012), comparisons with these forms are warranted before assignment of the humerus to a basal ornithomimosaur like Harpymimus. NHRD-AP 2015.v.103.9 differs from all of these in lacking a moderately developed to well-developed and large deltopectoral crest (Clark, Maryańska & Barsbold, 2004; Makovicky & Norell, 2004; Norell & Makovicky, 2004; Osmólska, Currie & Barsbold, 2004). NHRD-AP 2015.v.103.9 additionally differs from the humeri of tyrannosauroids (Holtz, 2004) in having a slight cleft separate the humeral head from the lateral tubercle. Thus, NHRD-AP 2015.v.103.9 can be assigned to a basal ornithomimosaur. Measurements of the both the proximal and distal portions of the humerus can be found in Table 1.

Figure 1 Humerus of an Arundel ornithomimosaur.

Left humerus of an indeterminate ornithomimosaur NHRD-AP 2015.v.103.9 in dorsal (A), ventral (B), lateral (C), medial (D), proximal (E), and distal (F) views. Black arrows indicate cleft separating humeral head from lateral tubercle. Blue arrow indicates edge of deltopectoral crest. Scale bars = 10 mm.

Table 1 Measurements of Arundel ornithomimosaur elements.

Specimen	Proximodistal length (measured on lateral (L) or dorsal (D) side)	Dorsoventral height (measured on proximal side)	Mediolateral width (measured proximal side)	Dorsoventral width (measured on distal side)	Mediolateral width (measured distal side)	
NHRD-AP 2015.v.103.9 (proximal end)	101 mm (D)	51 mm	30 mm	N/A	N/A	
NHRD-AP 2015.v.103.9 (distal end)	70 mm (D)	N/A	N/A	25 mm	58 mm	
NHRD-AP 2014.5.196	47 mm (L)	20 mm	18 mm	N/A	N/A	
NHRD-AP 2016.s.503	77 mm (L)	30 mm	25 mm	N/A	N/A	
NHRD-AP 2014.s.195	53 mm (L)	25 mm	22 mm	N/A	N/A	
USNM PAL 529423 (cast)	55 mm (L)	23 mm	22 mm	N/A	N/A	
NHRD-AP 2014.s.198	48 mm (L)	25 mm	19 mm	N/A	N/A	
NHRD-AP 2014.s.197	50 mm (L)	25 mm	20 mm	N/A	N/A	
NHRD-AP 2016.v.1104	30 mm (L)	23 mm	13 mm	N/A	N/A	

NHRD-AP 2014.s.196 (Figs. 2A–2D) represents the complete manual ungual of an ornithomimosaur. The manual ungual is elongate and flattened, and there is no flexor tubercle present. A small raised area on the ventral surface of this element is asymmetrically positioned and made up of leftover sediment, rather than being a flexor tubercle. The lack of a flexor tubercle and the flattened state of the manual ungual distinguishes this manual ungual from the previously listed clades of theropod dinosaur (Clark, Maryańska & Barsbold, 2004; Holtz, 2004; Makovicky & Norell, 2004; Norell & Makovicky, 2004; Osmólska, Currie & Barsbold, 2004). Instead, these morphologies ally the specimen with the manual unguals of ornithomimosaurs (Makovicky, Kobayashi & Currie, 2004). The grooves for the claw sheath are poorly defined, and in proximal view the ungual is ovoid in form. This manual ungual is most similar among basal and intermediate ornithomimosaurs to manual unguals II-3 and III-4 of the African species Nqwebasaurus thwazi (Choiniere, Forster & de Klerk, 2012). However, this manual ungual is much less recurved or elongated than those of N. thwazi and lacks any flexor tubercle. Instead, NHRD-AP 2014.s.196 best resembles ornithomimid manual unguals, such as those of Gallimimus (Makovicky, Kobayashi & Currie, 2004). This element is not an ornithomimosaur pedal ungual because it lacks a flexor fossa on its ventral surface. Thus, because of the lack of a flexor tubercle and its relatively elongate form, NHRD-AP 2014.s.196 is tentatively placed within Ornithomimosauria indet. Measurements of this element can be found in Table 1.

Figure 2 Selected elements of Arundel ornithomimosaurs.

Manual ungual NHRD-AP 2014.s.196 in lateral (A), proximal (B), dorsal (C), and ventral (D) views. Caudal vertebra NHRD-AP 2016.s.503 in lateral (E) and medial (F) views. Scale bars = 10 mm.

A caudal vertebra NHRD-AP 2016.s.503 (Figs. 2E and 2F) is mentioned here due to its similarities with the vertebrae described by Gilmore (1920) and its possible assignment to Ornithomimosauria indet. This elongate caudal vertebral centrum is somewhat similar to the caudal vertebrae described by Gilmore (1920), and as in the caudal vertebrae of ornithomimosaurs, the vertebrae is hollow (Buffetaut, Suteethorn & Tong, 2009). The centrum is ovoid in proximal view, and a portion of matrix obscures one end of the element. In some areas, the centrum is eroded. The dimensions of this vertebra are cataloged in Table 1.

Five pedal unguals of two different morphotypes were examined and provide the best evidence for the presence of two distinct species of ornithomimosaurs in the Arundel Clay ecosystem. These pedal unguals (Figs. 3A–3J and 4A–4O) share the presence of a flexor fossa on the ventral surface of each pedal ungual, the presence of relatively straight ventromedial edges on each of the unguals, and the presence of ventrolateral and ventromedial edges developed into keels, all diagnostic of ornithomimosaurs (Barsbold & Osmólska, 1990; Longrich, 2008; Makovicky et al., 2009; Xu et al., 2011; Choiniere, Forster & de Klerk, 2012; Lee et al., 2014). Though Choiniere, Forster & de Klerk (2012) noted that flattened pedal unguals and the presence of a flexor fossa on the ventral face of each pedal ungual were not exclusive traits to ornithomimosaurs, the two traits are mutually exclusive of other taxa. This is because flattened pedal unguals are only known in parvicursorine alvarezsauroids and Avimimus portentosus outside of Ornithomimosauria, whilst the presence of a flexor fossa on the ventral surface of each pedal ungual is only known in the abelisaurid Majungasaurus crenatissimus. The ventrolateral and ventromedial edges have been worn down in NHRD-AP 2014.s.197 and NHRD-AP 2016.v.1104. In NHRD-AP 2014.s.197, the flexor fossa on the ventral surface has been obscured by matrix. However, additional support for the placement of specimens NHRD-AP 2014.s.197, NHRD-AP 2014.s.198, and NHRD-AP 2016.v.1104 within Ornithomimosauria comes from their triangular shape in proximal view (Figs. 4E, 4J, and 4O), a trait considered a synapomorphy of Ornithomimidae by Barsbold & Osmólska (1990) and noted as a morphology found in Ornithomimosauria generally by Makovicky, Kobayashi & Currie (2004). Table 1 includes the measurements of these elements.

Figure 3 Pedal unguals of Arundel ornithomimosaurs.

Pedal ungual NHRD-AP 2014.s.195 in lateral (A), medial (B), dorsal (C), ventral (D), and proximal (E) views. Pedal ungual USNM PAL 529423 in lateral (F), medial (G), dorsal (H), ventral (I), and proximal (J) views. Black arrows indicate flexor fossa. Blue arrows indicate depression proximal to groove for claw sheath. Green arrows indicate ventrolateral and ventromedial keels. Scale bars = 10 mm.

Figure 4 Pedal unguals of Arundel ornithomimosaurs.

Pedal ungual NHRD-AP 2014.s.198 in lateral (A), medial (B), dorsal (C), ventral (D), and proximal (E) views. Pedal ungual NHRD-AP 2014.s.197 in lateral (F), medial (G), dorsal (H), ventral (I), and proximal (J) views. Pedal ungual NHRD-AP 2016.v.1104 in lateral (K), medial (L), dorsal (M), ventral (N), and proximal (O) views. Scale bars = 10 mm.

NHRD-AP 2014.s.195 is a slightly recurved and relatively elongate pedal ungual (Figs. 3A–3E; Table 1) with a proximal surface that is isosceles trapezoid-shaped and thins dorsally toward a proximodorsal process prominent in lateral and medial views. As in N. thwazi, NHRD-AP 2014.s.195 bears a well-defined flexor fossa with striations on its ventral surface (Choiniere, Forster & de Klerk, 2012). The center of NHRD-AP 2014.s.195 bears well-defined and deepened grooves for the claw sheath, a feature found in other ornithomimosaur pedal unguals (Makovicky, Kobayashi & Currie, 2004). The tip of NHRD-AP 2014.s.195 is blunt, likely due to erosion. The ventrolateral and ventromedial edges are developed into keels. In lateral and medial view, a depression sits directly proximal to the proximal end of each side’s groove for the claw sheath. In dorsal view, this feature creates a heightened ridge of bone that ends proximally in the proximodorsal process.

A cast of USNM PAL 529423 was also available for study (Figs. 3F–3J). USNM PAL 529423 is very similar to NHRD-AP 2014.s.195 in being relatively elongate (Table 1), having well-defined and deepened grooves for the claw sheath, and having a similar shape in proximal view. USNM PAL 529423 has an eroded ventral surface, but a weakly defined flexor fossa is still present. The flexor fossa seems to contain striations and is of a similar shape to that of NHRD-AP 2014.s.195 in being relatively circular. The ungual is very slightly recurved and has a blunt tip. The dimensions of USNM PAL 529423 compare closely with NHRD-AP 2014.s.195, though USNM PAL 529423 is 2 mm shorter than NHRD-AP 2014.s.195 (Table 1). This difference in height, however, seems less of a morphological difference than a taphonomic one. This is because the dorsal lip of the proximal face has been shortened by erosion (Figs. 3F–3J). Thus, the two millimeter difference is regarded here as a product of taphonomy. As in NHRD-AP 2014.s.195, a depression sits directly proximal to each of the grooves for the claw sheath in lateral and medial view. These depressions help to define a heightened ridge of bone in dorsal view that develops into the proximodorsal process.

NHRD-AP 2016.v.1104 is the proximal end of a pedal ungual (Figs. 4A–4E). This element has a somewhat flattened ventral surface in lateral and medial views and deviates from the morphology of NHRD-AP 2014.s.195 and USNM PAL 529423 in being triangular in proximal view. The grooves for the claw sheath are weakly defined. There is no indication of mediolateral curvature in ventral view. The ventrolateral and ventromedial edges of the specimen do not take the form of keels, though this may be due to erosion. NHRD-AP 2016.v.1104 also lacks depressions directly proximal to its grooves for the claw sheath.

NHRD-AP 2014.s.198 is a well-preserved recurved pedal ungual (Figs. 4F–4J). NHRD-AP 2014.s.198 is not as long proximodistally or nearly as wide mediolaterally as NHRD-AP 2014.s.195 or USNM PAL 529423. However, NHRD-AP 2014.s.198 is as tall as or taller dorsoventrally than NHRD-AP 2014.s.195 or USNM PAL 529423. This gives NHRD-AP 2014.s.198 a blunt appearance in comparison to NHRD AP 2014.s.195 and USNM PAL 529423 in lateral and medial view. The grooves for the claw sheath are poorly defined, dorsoventrally widened, and shallower than those of NHRD-AP 2014.s.195 and USNM PAL 529423. In proximal view, the pedal ungual is triangular in shape as in NHRD-AP 2016.v.1104. The tip is slightly worn, and a portion of the ungual is obscured by matrix. The ventrolateral edge is broken off, and the ventromedial edge is slightly worn. However, the ventromedial edge is still complete enough to show it had been developed into a keel. A flexor fossa is still present on the ventral surface of this specimen. In ventral view, the morphology of the claw indicates it was the pedal ungual of either pedal digit II or IV as it is curved mediolaterally. There are no depressions proximal to the grooves for the claw sheath in NHRD-AP 2014.s.198. In NHRD-AP 2014.s.198, the proximodorsal process is also rather weakly defined. Additionally, the flexor fossa of NHRD-AP 2014.s.198 lacks striations.

NHRD-AP 2014.s.197 (Figs. 4K–4O) is among the best preserved of the pedal unguals described herein. This pedal ungual is relatively blunt with poorly defined grooves for the claw sheath. Like NHRD-AP 2014.s.198, this pedal ungual curves mediolaterally in ventral view. The dimensions of NHRD-AP 2014.s.197 are very similar to those of NHRD-AP 2014.s.198 (Table 1), and therefore dissimilar to those of NHRD-AP 2014.s.195 and USNM PAL 529423. NHRD-AP 2014.s.197 is also slightly recurved to a similar degree as NHRD-AP 2014.s.198. As in NHRD-AP 2016.v.1104 and NHRD-AP 2014.s.198, NHRD-AP 2014.s.197 is semi-triangular in proximal view. As in NHRD-AP-AP 2014.s.198, NHRD-AP 2014.s.197 has dorsoventrally widened grooves for the claw sheath. A portion of matrix obscures a portion of the claw toward its distal end and the flexor fossa. The flexor fossa is obscured by matrix, and the proximodorsal process is weakly defined in NHRD-AP 2014.s.197. The ventromedial edge is worn, and the ventrolateral edge is obscured by matrix but present.

Discussion

Presence of two morphotypes of ornithomimosaur in the Arundel Clay

The Arundel ornithomimosaur material represents a significant record of one dinosaur group from the Early Cretaceous of Eastern North America. The described axial and appendicular material herein suggests that two different species of ornithomimosaur may have coexisted within the Arundel fauna. This is because of the presence of two different morphotypes of pedal unguals found at the Dinosaur Park site and because other ornithomimosaur specimens from the same site have affinities with more derived or more basal ornithomimosaur taxa. Both NHRD-AP 2014.s.195 and USNM PAL 529423 have well-defined, deepened grooves for the claw sheath, a proximal face that is isosceles trapezoid-shaped and thins dorsally to a prominent proximodorsal process, depressions proximal to their grooves for the claw sheath which contribute to the prominence of a raised portion of bone in dorsal view, and a flexor fossa with striations running through it on their ventral surfaces. NHRD-AP 2014.s.197 and NHRD-AP 2014.s.198 are triangular in proximal view, shorter and less widened mediolaterally than NHRD-AP 2014.s.195 and NHRD-AP 2014.s.197, have shallow grooves for the claw sheath, lack striations running through the flexor fossa on their ventral faces, and are more recurved than NHRD-AP 2014.s.195 and USNM PAL 529423. NHRD-AP 2016.v.1104 is also triangular in proximal view and lacks deepened grooves for the claw sheath. The questions of whether all the pedal unguals represent different ontogenetic stages of ornithomimosaurs or that the unguals are from different digits of the pes of an ornithomimosaur are also addressed here. Intraspecific and ontogenetic variation can be regarded as unlikely due to the disparate morphologies of the pedal unguals in proximal view and the lack of intermediate morphological traits between the morphotype represented by NHRD-AP 2014.s.197, NHRD-AP 2014.s.198, and NHRD-AP 2016.v.1104 and the morphotype represented by NHRD-AP 2014.s.195 and USNM PAL 529423. Further evidence against ontogenetic or intraspecific variation among these unguals comes from previously described unguals from the Arundel Clay assigned to Ornithomimosauria. Gilmore (1920) described a single pedal ungual, which he assigned to “Ornithomimus” affinis. The morphology of this ungual (USNM V 6107), which was more recently figured in Serrano-Brañas et al. (2016), closely corresponds to the pedal unguals NHRD-AP 2015.s.197 and NHRD-AP 2015.s.198 in being shortened, recurved, lacking striations in its flexor fossa, and lacking grooves for the claw sheath as well-defined and deep as those of NHRD-AP 2015.s.195 or USNM PAL 529423. Evidence against the claws being from different digits of the pes of a single species of ornithomimosaur stems from the significant differences in morphology between the unguals. In the ornithomimosaurs where the proximal faces of more than one pedal ungual is exposed and documented (e.g., Beishanlong grandis, Rativites evadens, Struthiomimus altus, Ornithomimus edmonticus, and Gallimimus bullatus), all pedal unguals are flattened or slightly recurved to a similar degree and share a distinct triangular shape in cross-section (Makovicky, Kobayashi & Currie, 2004; Makovicky et al., 2009; McFeeters et al., 2016). Thus, it is very unlikely that the pedal unguals described herein belong to different digits of the pes of one ornithomimosaur, as they significantly vary in their shape in proximal view and display different degrees of curvature. USNM V 6107 is also straightened in ventral view (Fig. 8B in Serrano-Brañas et al., 2016), suggesting that it came from pedal digit III. Thus, the blunt morphotype of pedal ungual represented by NHRD-AP 2014.s.197, NHRD-AP 2014.s.198, NHRD-AP 2016.v.1104, and USNM V 6107 is both represented by mediolaterally curved and straightened forms, suggesting that they not only represent the pedal unguals of digits II and IV, but also of digit III.

Additional support for the two different morphs of ungual described herein representing two different species stems from their morphological similarities to the unguals of ornithomimosaurs of different phylogenetic positions. NHRD-AP 2014.s.195 and USNM PAL 529423 are more similar to the unguals of derived ornithomimosaurs and ornithomimids in being flattened, having a prominent ridge of bone oriented proximodistally in dorsal view, having depressions directly proximal to the proximal end of their grooves for the claw sheath, and being elongate (Barsbold & Osmólska, 1990; Makovicky, Kobayashi & Currie, 2004; Serrano-Brañas et al., 2016). NHRD-AP 2014.s.197, NHRD-AP 2014.s.198, and USNM V 6107 share similarities with basal ornithomimosaurs in being recurved and blunt (Makovicky, Kobayashi & Currie, 2004; Jin, Chen & Godefroit, 2012). These features also provide evidence against the notion of sexual variation among ornithomimosaurs contributing to the presence of two morphotypes of pedal unguals.

Furthermore, evidence for the presence of two different ornithomimosaur species in the Arundel Clay stems from the non-pedal ungual material of ornithomimosaurs known from the Arundel. The humerus described herein is similar to the humerus of Harpymimus in being thickened compared to those of most other ornithomimosaurs (Makovicky, Kobayashi & Currie, 2004), suggesting the humerus is from an ornithomimosaur similar to H. okladnikovi. However, the manual ungual described herein is allied with ornithomimids in completely lacking a flexor tubercle (Makovicky, Kobayashi & Currie, 2004).

Other remains assigned to ornithomimosaurs from the Arundel Clay follow this pattern of sharing features with derived or basal ornithomimosaurs. The partial metatarsal III USNM V 5684 described by Gilmore (1920) resembles the metatarsal III of ornithomimids like Struthiomimus and Gallimimus much more than it does to Harpymimus in thinning only to abruptly expand near its diaphysis (Fig. 6.5 in Makovicky, Kobayashi & Currie, 2004; Serrano-Brañas et al., 2016). The metatarsal III is not as robust as those of arctometatarsalian tyrannosaurs (Holtz, 2004), and the lack of known tyrannosaur material from North America during the Aptian–Albian (only known from possible teeth) (Zanno & Makovicky, 2011) is suggestive that this element came from a subarctometatarsalian or arctometatarsalian ornithomimosaur. This coexistence of more derived and more basal ornithomimosaur species also occurs within the Yixian Formation of China (Ji et al., 2003; Jin, Chen & Godefroit, 2012).

Review of previously described Arundel Clay ornithomimosaur material

With the presence of two different species of indeterminate ornithomimosaurs within the Arundel Clay, a taxonomic reevaluation for the ornithomimosaur material of the Arundel is warranted. The material originally described by Lull (1911) as “Dryosaurus” grandis was placed within the ornithomimosaur genus Ornithomimus as “O.” affinis by Gilmore (1920). This material included dorsal and caudal vertebrae, a metatarsal II and metatarsal III, an astragalus, pedal phalanges, and a pedal ungual (Lull, 1911; Gilmore, 1920). Russell (1972) later placed the Arundel remains within Archaeornithomimus due to the curvature of the pedal ungual described by Lull (1911) and Gilmore (1920), while Smith & Galton (1990) regarded the material as indeterminate theropod remains. Serrano-Brañas et al. (2016) regarded the material as ornithomimosaur remains, using the pedal ungual USNM V 6107 and the metatarsal III USNM V 5684 in comparisons with other ornithomimosaur genera. Weishampel & Young (1996) and Weishampel (2006) listed other ornithomimosaur remains, including a tibia, as coming from the Arundel Clay. Indeed, multiple new specimens possibly belonging to ornithomimosaurs are currently in the collections of the National Museum of Natural History. However, as Gilmore (1920) does not specify the exact location of the pedal ungual and the metatarsal III relative to each other at the site near Muirkirk, Maryland which he gives as the location of their discovery and as the two different morphotypes of pedal unguals described herein come from the same site, it is best to conclude that all of the ornithomimosaur material previously described from the Arundel Clay may represent two different species and thus cannot be confidently assigned to one taxon.

Reassignment of N. justinhofmanni to Ornithomimosauria

Description of the new Arundel ornithomimosaur material and the naming of new basal ornithomimosaur taxa in recent years have allowed for the reinterpretation of the “generalized” North American coelurosaur N. justinhofmanni as an ornithomimosaur. This taxon, originally described as a coelurosaur of uncertain affinities, is known from three specimens from the Barremian Yellow Cat Member of the Cedar Mountain Formation (Kirkland et al., 1998). Nedcolbertia shares several synapomorphies with ornithomimosaurs and ornithomimids in having anteroposteriorly shortened phalanges from pedal digit IV, the ventral surfaces of the pedal ungual flattened in lateral view, being triangular in proximal view, having ventrolateral and ventromedial edges developed into keels, and having a flexor fossa on the proximal end of the ventral surface of its pedal unguals (Barsbold & Osmólska, 1990; Figs. 8 and 9 in Kirkland et al., 1998; Choiniere, Forster & de Klerk, 2012). The proximal end of metatarsal III is restricted mediolaterally in a very similar fashion to Harpymimus, and in proximal view the metatarsals are similar in shape to those of Kinnareeemimus and Nqwebasaurus (Fig. 8 in Buffetaut, Suteethorn & Tong, 2009; Choiniere, Forster & de Klerk, 2012). Additional features shared between Nedcolbertia and Ornithomimosauria include elongated caudal vertebra, an elongated femur with the femoral head directed straight medially and not separated from the greater trochanter by a sulcus, a large cnemial crest on the tibia which curves laterally toward the crest’s tip, elongated pedal unguals with deep grooves for the claw sheath, a tall ascending process on the astragalus, a complex space on the lateral side of the astragalus into which the calcaneum fits, a sulcus separating the astragalar condyles, and no sulcus separating the femoral head from the greater trochanter (Figs. 6C–6E, 8D–8N, and 9B in Kirkland et al., 1998; Makovicky, Kobayashi & Currie, 2004).

The subarctometatarsus of Nedcolbertia can be differentiated from that of tyrannosauroids by the proximally pinched metatarsal III not being limited to the plantar half of the foot (Holtz, 2004). Additionally, the metatarsal III of Nedcolbertia is not restricted to the extent seen in troodontids (Makovicky & Norell, 2004). Nedcolbertia can be differentiated from dromaeosaurids, troodontids, oviraptorosaurs, and therizinosaurs in lacking a moderately to well-developed deltopectoral crest (Clark, Maryańska & Barsbold, 2004; Makovicky & Norell, 2004; Norell & Makovicky, 2004; Osmólska, Currie & Barsbold, 2004). Unlike in tyrannosauroids, the greater trochanter of the femur of Nedcolbertia is not cleft from the femoral head.

Relationships of N. justinhofmanni and the Arundel Clay material

Nedcolbertia notably shares similarities in the morphology of its pes with a partial theropod pes from Arkansas that has been regarded as an ornithomimosaur (Quinn, 1973; Kirkland et al., 1998; Hunt-Foster, 2003). Kirkland et al. (1998) also related the Nedcolbertia material to the Arundel ornithomimosaur material, suggesting that alongside the Arkansas pes they may represent a distinct radiation. Unlike some ornithomimosaurs, the flexor tubercle of the first manual ungual is extremely pronounced, the manual unguals are likely differentiated, and the dorsal vertebrae are pneumatic with simplistic air sacs in Nedcolbertia (Kirkland et al., 1998). However, the morphology of the manual unguals of Nedcolbertia is indeed similar to the basal ornithomimosaurs Nqwebasaurus and Harpymimus in having large flexor tubercles. Additionally, Nedcolbertia and Nqwebasaurus can be linked by having slender pedal unguals, though this is less pronounced in the former taxon. The astragali are also morphologically similar in N. justinhofmanni and N. thwazi in the relative height of their ascending processes. Additionally, N. justinhofmanni can be distinguished from the derived Arundel ornithomimosaur morphotype in the morphology of their metatarsus. The known portion of the metatarsal III described by Lull (1911) and Gilmore (1920) suggests that at least one ornithomimosaur taxon found at the Arundel had a subarctometatarsalian condition similar to that of Kinnareemimus and derived ornithomimosaurs (e.g. Makovicky, Kobayashi & Currie, 2004; Buffetaut, Suteethorn & Tong, 2009; Claessens & Loewen, 2015), while in Nedcolbertia the dorsal face of metatarsal III is still completely visible along the entire portion of the metatarsus. Additional differences between the derived Arundel ornithomimosaur material and Nedcolbertia include the presence of noticeable flexor tubercles on the manual unguals of the latter taxon and the slightly more recurved nature of the derived Arundel ornithomimosaur’s pedal unguals in comparison with those of Nedcolbertia. Nedcolbertia can be distinguished from the derived morphotype of pedal ungual described herein by the presence of a ridge of bone separating the flexor fossa in two in the former taxon. Nedcolbertia can be distinguished from the more basal Arundel form in the elongate nature of the unguals of the former species (Figs. 8M–8N in Kirkland et al., 1998). Nedcolbertia is therefore considered as a basal ornithomimosaur due to its non-arctometatarsalian pedal condition where the shaft of metatarsal III is at least partially visible along its entire run in dorsal view.

Biogeographic implications

Along with the basal Arundel ornithomimosaur material and the unnamed pes from the Trinity Group of Arkansas, the presence of Nedcolbertia in the Cedar Mountain Formation shows that ornithomimosaurs were present across the continent of North America during the Early Cretaceous. The presence of a derived ornithomimosaur in the Arundel Clay also suggests that more derived ornithomimosaurs coexisted with these more basal forms. Ornithomimosaurs with derived traits, such as the loss of teeth in the dentary, already occur in Europe by the Hauterivian to Barremian (Allain et al., 2014). The presence of a species of basal ornithomimosaur in North America during the Barremian (Yellow Cat Member of Cedar Mountain Formation) (Kirkland et al., 1998) evinces that the clade had already spread to North America in addition to Africa, Europe, and Asia during the earliest stages of the Cretaceous (Allain et al., 2014). Allain et al. (2014) noted that the close relationships between the African taxon Nqwebasaurus and other basal ornithomimosaurs suggested that the group was widespread before the breakup of Pangaea. The North American ornithomimosaur record (Nedcolbertia, Arundel taxa, and Arkansas pes) may also indicate this. However, it is also a possibility that, as Allain et al. (2014) discussed, a European–Asian interchange resulting from low sea levels could have allowed the immigration of ornithomimosaurs to North America.

The presence of a large number of species of basal ornithomimosaurs (e.g., Nedcolbertia, Nqwebasarus, Hexing, and Pelecanimimus) (Choiniere, Forster & de Klerk, 2012; Jin, Chen & Godefroit, 2012) in North America, Asia, Africa, and Europe during the Barremian and earlier stages of the Cretaceous with little overlap of forms with more derived affinities (represented by an unnamed form from Angeac, France and possibly the British taxon Valdoraptor) suggests that basal ornithomimosaurs were the most common and widespread of ornithomimosaurs during the earliest Cretaceous. However, it is notable that toward the end of the Early Cretaceous (Aptian–Albian), these basal forms start to disappear as more derived taxa (including ornithomimids and deinocheirids) (e.g., Harpymimus, Beishanlong, an unnamed ornithomimid pes from China, the derived Arundel taxon) occur more often (Shapiro et al., 2003; Jin, Chen & Godefroit, 2012; Allain et al., 2014). Ornithomimosaur taxa are also known from only North America and Asia at this time and into the Late Cretaceous (Allain et al., 2014). Thus, it may be that a decrease in basal taxa coincided with the regression of the range enjoyed by ornithomimosaurs in the earliest Cretaceous. However, further discoveries of ornithomimosaurs from the Early Cretaceous are needed before this hypothesis can be thoroughly tested.

Conclusions

The Arundel ornithomimosaurs are some of the best-characterized theropods from the Early Cretaceous of Eastern North America. Furthermore, the Arundel material along with recently described ornithomimosaurs have provided evidence for the placement of N. justinhofmanni as an ornithomimosaur. Both these taxa and an unnamed ornithomimosaur pes from the Early Cretaceous of Arkansas affirm that ornithomimosaurs were present across North America during the Early Cretaceous.

Previous studies on ornithomimosaur remains have affirmed that individual postcranial elements assignable to Dinosauria can be assigned to family level and genus level (Currie, 1987; Longrich, 2008). Notably, Longrich (2008) and Shapiro et al. (2003) demonstrated that manual and pedal material can be diagnostic for Ornithomimosauria. The species Tototlmimus packardensis was recently named on the basis of solely manual and pedal elements (Serrano-Brañas et al., 2016). Thus, because Nedcolbertia and the Arundel material share a significant number of diagnostic traits with Ornithomimosauria in their manus and pes, both N. justinhofmanni and the ornithomimosaur material from the Arundel Clay can be assigned to Ornithomimosauria.

As Longrich (2008) stated, the importance of dissociated elements within an assemblage should not be understated. This is especially true for dinosaur specimens from the Eastern United States, which are often found dissociated and incomplete (Weishampel & Young, 1996). Here, inspection of the pedal unguals described and the hypothesis that they represent different basal and derived taxa is also consistent with the morphologies of other ornithomimosaur specimens from the Arundel Clay.

Both the Arundel material and Nedcolbertia have implications for the biogeography of Early Cretaceous ornithomimosaur lineages, suggesting that basal and more derived ornithomimosaurs had a wide biogeographic range during the Early Cretaceous and may have coexisted in at least Asia and North America. Additionally, the interpretation of the Arundel Clay material representing two different taxa suggests similarities between the dinosaur fauna of the Eastern United States and the Yixian Formation of China (Ji et al., 2003; Jin, Chen & Godefroit, 2012). Continued study of the often-dissociated dinosaur specimens of the Arundel Clay is likely to aid in the understanding of the biodiversity of dinosaurs in the Eastern United States during the Early Cretaceous and provide further data on the evolution of ornithomimosaurs and other smaller theropod groups during the Cretaceous as a whole.

The author would like to thank Benjamin Miller for allowing access to view the Dinosaur Park collections. Lastly, the researcher is thankful to Dr. Matthew Wedel, Dr. Thomas Holtz, and an anonymous reviewer for their helpful comments and suggestions.

Additional Information and Declarations

Competing Interests

Author Contributions

Data Deposition

The authors declare that they have no competing interests.

Chase Doran Brownstein conceived and designed the experiments, performed the experiments, analyzed the data, contributed reagents/materials/analysis tools, wrote the paper, prepared figures and/or tables, and reviewed drafts of the paper.

The following information was supplied regarding data availability:

The research in this article did not generate, collect or analyze any raw data or code.

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
