# Peer review of "Description of Arundel Clay ornithomimosaur material and a reinterpretation of Nedcolbertia justinhofmanni as an “Ostrich Dinosaur”: biogeographic implications"

_PeerJ, doi:10.7717/peerj.3110_

## Round 0.1 · original submission · Major Revisions

With reference to our emails, I see three areas in which the manuscript can be usefully improved before it goes out for peer review:

1. Update the description as needed once you've had a chance to study the Arundel ornithomimosaur material firsthand.

2. At least briefly describe what equipment and software you used to take measurements and compose the figures ("I photographed the fossils using a Blahblah DSLR and cropped out the backgrounds in Photoshop CS" or whatever).

3. Organizing the reference section alphabetically by the authors' last names is standard at PeerJ, and doing so will make it a lot easier for the reviewers to use the reference section while reading the paper, and to double-check it against the citations in the text.

You may also want to include at least a small table of measurements, since scale bars in photographs are notoriously unreliable. To be clear, I'm not requiring you to do this at this stage, but it's good practice, and it's something the reviewers may well request anyway.

---

## Round 0.2 · Major Revisions

Despite the weaknesses of the manuscript in its current form, Reviewer 1 makes a good case in the first two paragraphs for seeing this descriptive work into publication. I am therefore offering you the opportunity to make major revisions to the manuscript, rather than rejecting it.

Concerns raised by the reviewers include (but are not limited to): (1) association of the material - does it all belong to one taxon, and if so, what evidence supports that conclusion; (2) additional possible identifications (e.g., basal tyrannosauroid) need to be tested; (3) the conclusion that the material represents two morphotypes - as opposed to lying along a spectrum of intraspecific variation, or simply representing different digits - is not well supported; and (4) the inference that the material belongs to Ornithomimosauria would be stronger if it was supported by a phylogenetic analysis.

Those are only the most major of the reviewers' concerns. I have read carefully though all of their suggestions and I find them all apt and well-reasoned, and I encourage you to address all of them in revising the manuscript.

The reviewers found the descriptive work less problematic, and the inferences regarding phylogenetic placement and morphotypes to be more so. In revising the manuscript, you might consider concentrating on the stronger, descriptive section and cutting the more speculative sections, unless you can successfully address the reviewers' concerns. A phylogenetic analysis would be a straightforward and valuable addition to the manuscript in any case.

·

Basic reporting

The overall manuscript is fine. It serves to do that vital task for the field of paleontology that journals like Science and Nature do not do: document the nitty gritty details of the partial specimens that make up the vast majority of the fossil record, and put these into a broader context in terms of phylogenetic and biogeographic distribution.

I would definitely like to see this manuscript (in a modified form) published, as the observations of the author will be important in sorting order out of the isolated material that characterizes the Arundel.

That said, I have some concerns about the strength of the primary conclusion of the paper; some minor issues with the stratigraphic nomenclature; and a few other corrections and comments. I do not see any of them as fatal to the paper, but I do think they should be considered before the final version is achieved.

The primary conclusion: The author’s primary conclusion is that the material is from two different ornithomimosaur taxa. I would agree that this is a possibility consistent with the evidence. However, I disagree that the author has actually sufficiently demonstrated this.

Firstly, the author does not seem to have tested an additional possible identity, and one which would be consistent with some of the specimens in terms of size and morphology: that these are from basal tyrannosauroids. Basal tyrannosauroids are known to be present from several formations of comparable age around the world (the Cloverly, the Wessex, the Yixian, etc.). I am not stating that these specimens definitely belong to tyrannosauroids, simply that the author must eliminate this possibility as they did for dromaeosaurids, troodontids, etc.

Furthermore, I would encourage caution here with regards to the assertion of two distinct taxa. The two different morphotypes of pedal unguals MAY indicate the presence of two different ornithomimosaur taxa. Let us keep in mind that without associated material, we can’t eliminate such possibilities as these representing different toes having different proportions, for instance. Indeed, I would say that should be null hypothesis (i.e., that they are unguals from different digits), which the author must reject. Other possibilities are ontogenetic or sexual dimorphic differences. I do agree that coupled with the metatarsal evidence presented there is certainly evidence to suggest two morphotypes, but caution should always be stated when dealing with fragmentary, disarticulated material.

Stratigraphic nomenclature: As might be predicted for units with extremely limited outcrop and consequently very limited ability to assess lateral continuity, there remains a lack of resolution in the nomenclature for the Arundel Clay unit. Traditionally it is considered a Formation, as used in the present manuscript. Others, however, consider it a Member within the Patuxent Formation (e.g., Stanford et al. 2010); still others include both the Patuxent and the Arundel as two of the Members of the Potomac Formation (e.g., Lipka et al. 2006). The author should at least address the existence of different nomenclatures, then state the version they will follow. In line 97 and following the author follows the Lipka model, but early in the manuscript refers to the “Arundel Formation” (which therefor contradicts the Lipka model).

As a further note: the term “Arundel facies”, while certainly legitimate in broad strokes, is not a formal term in lithostratigraphy. “Arundel Clay” (= “Arundel Formation”) or “Arundel Clay Member” (= “Arundel Member”) would be the proper terminology following US Geological Survey practices, depending on which particular scheme you follow.

Lipka, T.R., F. Therrien, D.B. Weishampel, H.A. Jamniczky, W.G. Joyce, M.W. Colbert & D.B. Brinkman. 2006. Journal of Vertebrate Paleontology 26: 300-307. DOI: 10.1671/0272-4634(2006)26[300:ANTFTA]2.0.CO;2
Stanford, R., R.E. Weems & M.G. Lockley. 2010 A new dinosaur ichnotaxon from the Lower Cretaceous Patuxent Formation of Maryland and Virginia. Ichnos 22: 251-259. DOI: 10.1080/10420940490428797

Experimental design

These were discussed in the preceding section, but just to reiterate them:

* The author must find a means to reject what I would consider the null hypothesis: that the different unguals represent different digits rather than different taxa. This might be done by comparing these to taxa in which we have different digits from the same specimen.

* The author must consider and find justification to reject a tyrannosauroid affinity for the specimens, as they have done for possible dromaeosaurid, troodontid, therizinosauroid, and oviraptorosaur affinities.

Validity of the findings

Again, this was also covered in the first section, but for clarity's sake:

It is by no means definitive that the differences between the individual unguals represent taxonomic differences, rather than anatomical ones.

(I do not actually disagree with the author per se, but they need to be able to reject this and other alternatives.)

Additional comments

Additional comments and corrections:
Line 45: A quibble and a complication. The quibble is the use of “new species” here: the point is that it would in fact be an old species! So a better phrase would be “determined to be a distinct and valid species”.
Furthermore, there is a taxonomic complication: The trivial nomen grandis would be the appropriate one for the taxon, except for the (admittedly unlikely) condition in which the Arundel form was found to lie within the Maastrichtian Laramidian genus Ornithomimus. That combination, Ornithomimus grandis, already exists and long pre-dates Lull’s form (Marsh 1890, the same paper in which Ornithomimus itself was proposed.) (“Ornithomimus grandis Marsh 1890” turns out to be undiagnostic tyrannosaurid material, but nevertheless occupies that name.)

Line 67 (also 143, 146, and throughout): Ornithomimosauria, a formal taxon name, must be capitalized.

Line 124: Capitalize the “Early” in “Early Cretaceous”.

Lines 133 ff: Correct “manal” to “manual”. Gauthier (1986) famously misspelled this word, and that misspelling has taken on a life of its own. But the adjectival form of “manus” is “manual”.

Line 148: As a formal name, “Theropoda” must be capitalized.

Line 195: Does it REALLY represent one of the most important records of an Early Cretaceous dinosaur group? Even as a theropod worker, I find that to be a statement that isn’t really that easy to back up. This discovery documents a group already known on many continents at that time, millions of years after the oldest known appearance of the clade, and not documenting any particularly unusual new morphology or phase in its evolutionary history. Significant, agreed.

Line 217: White (2009) dubbed this morphology the “subarctometatarsus”.

White, M.A. 2009. The subarctometatarsus: intermediate metatarsus architecture demonstrating the evolution of the arctometatarsus and advanced agility in theropod dinosaurs. Alcheringa 33: 1-21. DOI: 10.1080/03115510802618193

Line 279 (and elsewhere): When you run into a situation where you have two different genera with the same initial letter, it might be preferable to not simply abbreviate them in the same fashion. (Otherwise, it is temporarily misleading in suggesting they belong to the same genus.) One alternative that is used on occasion is to use the single letter abbreviation for one, and a two letter abbreviation for the other. Since Nqwebasaurus is likely the only word beginning with “nq” most English-readers are likely to encounter, I would suggest you refer to his taxon as Nq. thwazi and Nedcolbertia as N. justinhoffmani. Alternatively, as these are each the only species known in their respective genera (at present), you might just use the genus names.

Line 295: A note of caution: an adaptive radiation is a very particular thing, namely a diversification from a single ancestor not shared with other comparable taxa. You have not actually shown data to support this (such as a phylogenetic analysis wherein the Early Cretaceous ornithomimosaurs form their own clade). The shared presence of intermediate-grade morphologies does not necessarily point to a radiation. In particular, you would need to compare these Early Cretaceous forms to ornithomimosaur outgroups (basal maniraptorans, compsognathids, basal tyrannosauroids, etc.) to see if the traits the author suggests as evidence of a “radiation” are unique to these ornithomimosaurs, or simply the retention of primitive traits lost in derived ornithomimosaurs.

Line 338: I would suggest “best characterized” rather “well-known” here; they are rather obscure specimens so far, but the author has done much to detail their anatomy.

Reviewer 2 ·

Basic reporting

The English is overall proficient although there are occasional minor grammatical and spelling errors. I believe the literature references are adequate for the scope of the study. However, there are notable items that should be addressed:

1. The anatomical descriptions and figures are basic and do little to build on the work of previous studies on the same and related material (e.g., Gilmore 1920). For example, “the dorsal vertebrae are simplistic and pneumatic in Nedcolbertia” (Ln 274–275) without details on the “simplistic” and “pneumatic” features. For a compelling descriptive work, the author should provide improved photographs (e.g., better lighting to make specific characters easily visible), additional photographs that highlight the diagnostic features, and labels (e.g., features, anatomical directions, scale bars in all components of the figure).
2. Be consistent in the use of “ornithomimosaur” vs. “ornithomimosaurian” as adjective and noun. If the author intends different meanings, then please specify.
3. Ln 29: Please clarify why Arundel facies represent the “best” dinosaur fauna among the early Cretaceous or eastern US localities (e.g., number of taxa described?).
4. Ln 82–84: Typically, there is no need to describe permits or access if the examined specimens are only from collections (i.e., no new material from fieldwork).
5. Ln 87–89, 91–92: Please include city and state for these institutions and companies.
6. Change every instance of “manal” ungual to “manual.”
7. There were many minor, but notable, grammatical and spelling errors throughout the manuscript. It should be proofread at least once more for these errors. For example, “De Klerk” should be “de Klerk,” “Hasegwa & Manabe” should be “Hasegawa & Manabe.”

Experimental design

The study aims to describe and classify isolated axial and appendicular elements from the early Cretaceous Arundel facies. I agree with the author’s sentiment that the early Cretaceous Arundel fauna is crucial for filling both temporal and biogeographical gap in our understanding of North American dinosaurs. However, the study’s methodology is questionable at best:

1. Despite being of great importance to the study’s aim, the manuscript does not provide vital information on the discovery or association of the fossil specimens. For example, were the isolated elements discovered from the same site (including the USNM specimen)? This is necessary information also because after the author assigns the damaged humerus to Ornithomimosauria indet., the remaining elements are compared only to other ornithomimosaurian taxa as if they are assumed to be ornithomimosaurian in origin.
2. Central to the study’s primary objective is the definitive assignment of these specimens to Ornithomimosauria. However, the justifications for the taxonomic assignment are not rigorous and in some cases, not given. For instance, the anatomical comparisons to other ornithomimosaur taxa are based on visual similarities and without mention or reference to their diagnostic power (e.g., Ln 152–156: “flatness of some unguals,” “triangular shape of some [unguals] in proximal view”). Additionally, the explanation for assigning the isolated caudal vertebra to Theropoda indet. is not even mentioned (Ln 146–148).
3. Likewise, the claim for the existence of two divergent ornithomimosaurian taxa in the Arundel fauna is also poorly justified. First, this conclusion is made based on very limited set of isolated postcranial elements and primarily on purported existence of two “morphs” of pedal unguals. Secondly, the morphological characters used are dubious (e.g., “relatively less-elongate […] having dorsoventrally widened grooves for the claw sheath” (Ln 205–208). Thirdly, the study lacks any quantitative analysis that provides evidence of two distinct morphs (e.g., morphospace) rather than a continuum of potential intraspecific variation.
4. I would find it worthwhile to check if incorporating the Arundel specimens and Nedcolbertia into phylogenetic morphological characters will yield trees with these specimens placed within Ornithomimosauria. If the character state combinations stated in the study are indeed diagnostic, then the specimens will be placed accordingly in a parsimony tree despite high proportion of missing data.

Validity of the findings

I believe that the study does not contribute sufficient amount of novel information in its current state due to poorly described and figured specimens and claims that are not robustly supported. As stated in my previous comments, the assignment of these specimens collectively to Ornithomimosauria is speculative and not articulated in the manuscript. An exemplary passage of this is as follows, regarding the assignment of the humerus to Ornithomimosauria (Ln 122–127): “Because dromaeosaurids (e.g., Deinonychus), troodontids (e.g., Geminiraptor), oviraptorosaurs (=Microvenator), and therizinosaurs (e.g., Falcarius) are also known from the Arundel facies and other North American formations of similar early Cretaceous age […], comparisons with these forms are warranted before assignment of the humerus to a basal ornithomimosaur like Harpymimus. NHRD-AP 2015.v.103.9 differs from all of these in lacking a moderately developed to massive deltopectoral crest […]. Thus, NHRD-AP 2015.v.103.9 can be assigned to a basal ornithomimosaurian dinosaur.” In addition to not explaining how the author defines “massive,” this character is used despite the fact that the earlier sentence in the same paragraph, the author reports that the “deltopectoral crest is eroded” (Ln 117) making the interpretation dubious.

Another key conclusion of the study is that there are two morphs of ornithomimosaurians. As stated above in my review, I believe that this claim is speculative because (1) these specimens are dubiously placed within Ornithomimosauria; (2) the association between the specimens are not mentioned to show that these purported two morphs were co-eval; (3) quantitative analysis of two distinct morphs, instead of a spectrum of intraspecific variation, is lacking. For instance, there is no discussion on ontogenetic changes and size differences between the specimens. Evidence must be shown that these purportedly diagnostic features are robust to instraspecific, including ontogenetic, variation.

Similarly, the conclusion that Nedcolbertia justinhofmanni is an ornithomimosaurian is interesting to a theropod specialist, but lacks rigor expected from a published study on this topic. The study mentions few synapomorphies seen in published figure of Nedcolbertia (Kirkland et al. 1998) attributable to Ornithomimosauria (Ln 250–259), but none are exclusive features of the clade as stated in Choiniere et al. 2012, p. 12. I believe that the claim would be far more convincing if the specimens were scored using the current set of morphological characters especially since, as the author states, there have been discoveries of basal ornithomimosaurian members in recent years that may help place Nedcolbertia under a phylogenetic analysis.

Additional comments

To make the study compelling, I recommend:
• Improving the anatomical description with additional detail and improved figures that highlight the key features. In my opinion, providing an illustration of the specimens that clarify the anatomical features would be very helpful to the readers for verifying the author’s claims.
• Providing information on the association between the specimens.
• Providing clear justifications for (1) assigning each element to Ornithomimosauria indet. using definitive, diagnostic features unique to Ornithomimosauria; (2) the co-occurrence of two divergent ornithomimosaur taxa with evidence from quantitative analyses; and (3) more comprehensive comparative analysis of Nedcolbertia.

---

## Round 0.3 · accepted · Accept

Thanks for your diligence in revising the manuscript.

The decision of whether or not to publish the peer reviews alongside the paper is entirely yours, and will not affect how your paper is handled going forward. However, I encourage you to do so. Making the reviews public allows the reviewers to receive more credit for their efforts, and also contributes to the emerging culture of fairness and transparency in editing and peer review.

·

Basic reporting

I consider all the previous objections/concerns of myself and the other reviewer as having been sufficiently dealt with, and would agree that the manuscript is now publishable as it stands.

Experimental design

I consider all the previous objections/concerns of myself and the other reviewer as having been sufficiently dealt with, and would agree that the manuscript is now publishable as it stands.

Validity of the findings

I consider all the previous objections/concerns of myself and the other reviewer as having been sufficiently dealt with, and would agree that the manuscript is now publishable as it stands.

Additional comments

I consider all the previous objections/concerns of myself and the other reviewer as having been sufficiently dealt with, and would agree that the manuscript is now publishable as it stands.